# Less is More – Towards parsimonious multi-task models using structured sparsity

Richa Upadhyay[1], Ronald Phlypo[2], Rajkumar Saini[1], Marcus Liwicki[1]

[1]Luleå University of Technology, Sweden, [2]University Grenoble Alpes, France

`richa.upadhyay@ltu.se, ronald.phlypo@grenoble-inp.fr, rajkumar.saini@ltu,se,`
`marcus.liwicki@ltu.se`

Model sparsification in deep learning promotes simpler, more interpretable models with fewer parameters. This not only reduces the model's memory footprint and computational needs but also shortens inference time. This work focuses on creating sparse models optimized for multiple tasks with fewer parameters. These parsimonious models also possess the potential to match or outperform dense models in terms of performance. In this work, we introduce channel-wise $l_1/l_2$ group sparsity in the shared convolutional layers parameters (or weights) of the multi-task learning model. This approach facilitates the removal of extraneous groups i.e., channels (due to $l_1$ regularization) and also imposes a penalty on the weights, further enhancing the learning efficiency for all tasks (due to $l_2$ regularization). We analyzed the results of group sparsity in both single-task and multi-task settings on two widely-used Multi-Task Learning (MTL) datasets: NYU-v2 and CelebAMask-HQ. On both datasets, which consist of three different computer vision tasks each, multi-task models with approximately 70% sparsity outperform their dense equivalents. We also investigate how changing the degree of sparsification influences the model's performance, the overall sparsity percentage, the patterns of sparsity, and the inference time.

## 1. Introduction

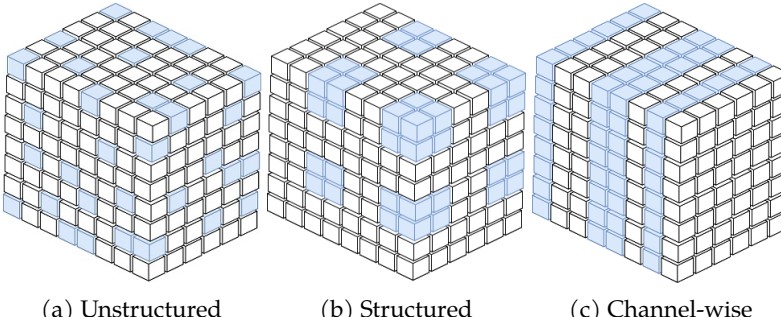

(a) Unstructured  (b) Structured  (c) Channel-wise

Figure 1: An illustration of the various forms of sparsity that may be introduced to a parameter vector of a Convolutional Neural Network (CNN). (blue denotes non-zero values). Figure (a) depicts unstructured sparsity, (b) shows structured (block or group) sparsity, and (c) represents channel-wise structured (group) sparsity.

The principle of parsimony states that a model with a lower number of parameters is preferred over a more intricate model with a higher number of parameters, given that both models fit the data equally well [1]. Model sparsification, one of the methods to obtain parsimonious models and model compression, holds significant importance in the fields of Machine Learning (ML) and Deep Learning (DL), often being explored through feature or parameter selection techniques. Although, in recent times, the concept of over-parameterization [2] i.e., model having more parameters than necessary to fit its training data and DL have become intertwined [3]. The pursuit of sparse models does not necessarily conflict with the concept of over-parameterization. In large-scale, complex scenarios, the benefit of sparse models is manifold. They provide increased interpretability, reduce overfitting, efficient computation, and aid in identifying the most informative features, leading to an efficient learning process [4]. While over-parameterization enables networks to approximate complex mappings and simplify the loss function, making optimization easier, model sparsification provides a different perspective that prioritizes efficiency and interpretability, both of which can be critical in specific applications. The contrast between the efficiency of sparse models and the comprehensive functionality of over-parameterized models showcases the continuous evolution

First Conference on Parsimony and Learning (CPAL 2024).

of DL. Continuing along these lines, our work delves into the adoption of structured sparsity to attain parsimonious models. We focus on identifying and leveraging the most significant features or parameters of a model, thereby balancing the efficiency of sparse models with the vast capabilities of deep (over-parameterized) models.

Another technique for reducing model size is parameter sharing, typically utilized in the context of MTL. Sparsity within an MTL framework, which seeks to train multiple tasks concurrently [5], can yield significant advantages. It is because, in MTL, where the model complexity increases as the number of tasks increases, a simple (sparse) model can be more interpretable and computationally efficient. Moreover, only some features might be relevant to all the tasks and oversharing of knowledge (features) among the tasks may cause negative information transfer [6]. As a result, sparse models can help to choose significant features for specific tasks while also assisting in learning shared representations across multiple tasks. Therefore, the motivation for this study is to leverage the advantages of structured sparsity to optimize MTL models.

This work, therefore, introduces structured (group) sparsity in MTL, i.e., it learns sparse shared features among multiple tasks. There are two important reasons for this integration. First is the organization of parameters in a CNN is inherently grouped into layers, channels, or filters; these naturally grouped parameters provide an opportunity to apply structured sparsity [7]. Another reason is CNNs, especially deeper ones, can develop redundant filters that extract similar features [8, 9]. This is because the number of filters in deep CNNs is usually thousands, and it is inevitable that there exist a lot of similar filters that extract the same or similar features [8]. Structured sparsity optimizes these networks by targeting and pruning entire redundant filters or channels of the weight matrix rather than just individual weights. Figure 1 illustrates the types of sparsity induced in a 4D parameter tensor (number of filters × channels × height × width). In this work, we introduce channel-wise group sparsity in the shared network among all the tasks, eventually eliminating a significant amount of shared parameters (i.e., groups or channels zeroed out). As a result, decreasing the memory footprint of the model will lead to less computational expense at the time of inference. Our approach is based on the understanding that the dense prediction tasks vary in the granularity and type of features they require. Some tasks might necessitate more low-level features (e.g., edge or texture details for semantic segmentation) while others could demand more high-level representations (like object or scene understanding for depth estimation). Channel-wise sparsity may allow for the retention of task-specific channels that are most pertinent while pruning the less relevant ones. The main contributions of this paper are as follows:

1. Introducing structured (group) sparsity in an MTL framework, particularly channel-wise $l_1/l_2$ penalty to the shared (CNN) layer parameters to solve complex computer vision tasks. Along with reducing the number of shared parameters, it aids in significantly improving task performance.

2. Design of an experiment framework to analyze the effect of group sparsity in single-task and multi-task settings.

3. A deep analysis of the model's performance across different levels of sparsification, taking into account the performance of the tasks, group sparsity, parameter reduction, and inference time.

4. A comparative analysis of the structured ($l_1/l_2$) vs unstructured ($l_1$) sparsity.

This article is organized as follows: The research works that used structured sparsity in the context of MTL are discussed in Section 2. Section 3 introduces multi-task learning, the concept of $l_1/l_2$ group sparsity, and presents the approach proposed in this work. Section 4 gives a detailed experimental set-up, while the results, followed by an extensive performance analysis, are presented in Section 5. At last, Section 6 draws the conclusions and future work.

## 2. Related work

Parameter-sharing in MTL enables models to exploit similarities across various tasks, enhancing generalization and learning efficiency. There are two types [5]: hard parameter-sharing and soft parameter-sharing. Hard parameter-sharing is caused by the network design, which shares some parameters across all tasks [10–12]. In contrast, soft parameter-sharing encourages models to have comparable but distinct parameters for the shared layers by including a regularization term in the loss function that penalizes the variations in the parameters [13]. However, as tasks increase,

the parameters and, consequently, the computational cost grow proportionally. Notably, not all tasks require complex networks and static architectures might lead to suboptimal results and often overfitting. Numerous studies in the past have optimized multi-task network structures for each task to overcome these issues. In this section, we review recent studies emphasizing parameter efficiency in MTL. We also explore the literature on structured sparsity both in general contexts and specifically within MTL.

**Parameter-efficient MTL:** MTL heavily uses Neural Architecture Search (NAS) [14] to optimize network design based on task relatedness and complexity of numerous tasks. Most pruning solutions start with a sizeable multi-task network and then propose a layer-wise architecture search process. [15] tailored residual networks to develop a task-specific ensemble of sub-networks in different depths, which are learned based on task similarities. Other approaches begin with a small network and incrementally expand the architecture, like [16], which starts with a thin multi-layer network and dynamically uses greedy algorithms to widen it during training. In their work, [17] proposed utilizing a differentiable tree-structured topological space to optimize parameters and branching distributions that enable automatic search of a hard parameter-sharing multi-task network. A method for learning the layer-sharing pattern over several tasks is proposed in [18], which involves first learning a task-specific policy distribution and then randomly selecting a select-or-skip policy decision from that distribution. Several other works, such as [19–21], propose different approaches to learning task-specific sparsity inducing 'masks' for the parameter vectors, which help to reduce the number of trainable parameters in a multi-task setting.

**Concept of parameter sparsity in DL:** [7] is a very detailed review that discusses the types of sparsity and their application in DL. Broadly, they categorize sparsity into two types: model sparsity, which involves pruning weights or neurons, and ephemeral sparsity, which is applied on a per-instance basis and deactivates neurons or weights, such as with dropout or activation functions. Numerous studies in the literature have adapted these sparsification methods for various applications; we highlight a few of these, particularly ones involving structured sparsity. The work presented in [22]) introduced block-sparse regularization in the form of $l_1/l_q$ norms [23] to acquire a low-dimensional representation that can be shared across a group of related tasks. They proved that penalizing the trace norm of the parameter matrix to make it low rank makes the non-convex group lasso optimization problem convex. A combination of block-sparse and element-wise sparse regularization is proposed in [24] to improve the block-sparse models since their performance depends on the extent to which features are shared across tasks. Similarly, [25] introduced channel-wise, stripe-wise, and group-wise filter pruning as a means of introducing sparsity in Deep Neural Network (DNN). The issue of addressing the estimation of multiple linear regression equations and variable selections within a multi-task framework is discussed in [26]. Therefore, many studies utilize structured sparsity in the context of MTL, focusing on variable selection, identifying task-specific layers, and learning sparse binary masks for the parameters of each task.

**Positioning our work:** While most of the works in the field of MTL employ the concept of sparsity for feature selection and usually use synthetic data for simple regression tasks [22, 24, 26], and learn sparse (unstructured) masks for individual tasks [19]; our work employs model sparsity and ephemeral sparsity while training a multi-task model for various heterogeneous tasks [27], such as image-level tasks (e.g., classification) and pixel-level tasks (e.g., segmentation, depth estimation). [28] is very close to what we have done; however, they propose a fusion regularization term in their work for task grouping and demonstrate the performance on synthetic data. Our primary contribution lies not in the introduction of sparsity itself; instead, it centres on the investigation and applicability of the group sparsity in MTL for dense prediction heterogeneous computer vision tasks. In the context of this work, model sparsity is applied in the form of channel-wise $l_1/l_2$ regularization as shown in Figure 2, which involves pruning of shared model weights during training, it impacts not only the forward pass of the multi-task model but also its inference in terms of performance, complexity, and prediction time. The shared CNN (i.e., ResNet, discussed in Section 4) has ReLU activation functions after every convolutional layer. This naturally introduces ephemeral sparsity per data point during training, which only impacts the training stage [7]. Consequently, in this study, we concentrate solely on the effects of model sparsity. It is important to note that our work centres on the sparsification of hard-shared parameters, specifically the backbone parameters typically shared by all tasks. However, this concept can also be adapted for the soft-parameter sharing setting.

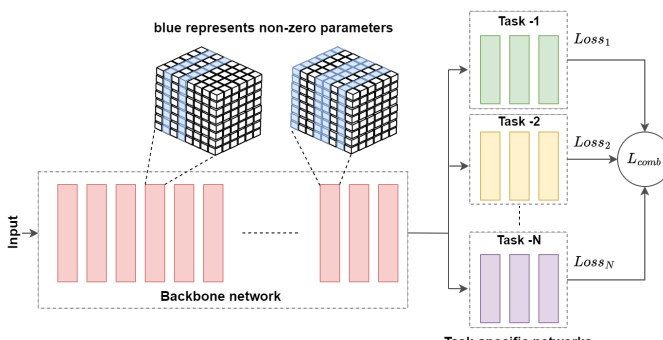

Figure 2: This block diagram is a simple illustration of the proposed work. Here, all the shared network layers aim for sparsity, as their channel-wise parameters are grouped and subjected to the $l_1/l_2$ penalty. The objective is to learn sparse multi-task shared representations that help to enhance performance across all tasks. Simultaneously, the task-specific networks focus on learning representations tailored for each individual task. This multi-task network is trained using backpropagation on a composite loss derived from the individual losses of each task.

## 3. Methodology

In this section, we begin by establishing the foundational concepts of our work, delving into MTL and group sparsity-inducing penalties. Subsequently, we introduce our proposed approach.

**Multi-task learning (MTL) :** MTL prioritizes the simultaneous training of multiple tasks instead of isolated training. This approach leverages common knowledge and representations across multiple tasks, often leading to enhanced performance and generalization [27, 29]. Consider a simple multi-task architecture that demonstrates hard parameter sharing as in Figure 2, that shares a common backbone network between many tasks and each task having a separate task-specific network. For $N$ non-identical but related tasks sampled from a task distribution $\mathcal{T} = \{T_i\}_{i=1}^{N}$, let $\theta_b$ be the shared parameters of the backbone network (shared layers), while $\{\theta_i\}_{i=1}^{N}$ be the task-specific parameters for $N$ tasks, such that $\theta_b \cap \{\theta_i\}_{i=1}^{N} = \emptyset$. In MTL, the objective is to minimize the combined loss $L_{comb}$ of all the tasks by finding the optimal network parameters $\theta = \theta_b \cup \{\theta_i\}_{i=1}^{N}$. It can be expressed as:

$$\theta^* = \arg\min_{\theta} L_{comb}(\theta, D^{tr}), \tag{1}$$

where $D^{tr} = \{D_i^{tr}\}_{i=1}^{N}$ represents the training dataset of $N$ tasks. Consider a function $\mathcal{F}$ that illustrates how the losses from all the tasks are combined for being back-propagated to the multi-task network, i.e., $L_{comb} = \mathcal{F}(L_1, L_2, .., L_N)$. Various techniques for aggregating the losses across multiple tasks have been discussed in [30]. This work employs the concept of uncertainty weighing [31] to balance the multiple losses. It can be mathematically represented as:

$$L_{comb} = \sum_{i=1}^{N} \left( \frac{1}{2\sigma_i^2} L_i(\theta_b, \theta_i, D_i^{tr}) + log(\sigma_i) \right), \tag{2}$$

where $\{\sigma_i\}_{i=1}^{N}$ are learnable parameters that are optimized along with the model parameters $\theta$ to minimize the combined loss.

**Structured sparsity inducing penalty:** The group lasso penalty [32], which combines the $l_1$ and $l_2$ norms (hereafter referred to as $l_1/l_2$ in this manuscript), is a form of regularization that induces structured sparsity [33]. Its objective is to generate solutions that eliminate entire groups of variables, in contrast to the most frequently used $l_1$ norm, which results in unstructured sparsity. The group $l_1/l_2$ penalty is introduced as a regularization term along with the loss function; the optimization objective is then expressed as:

$$\min_{\theta} \ L(\theta, D^{tr}) + R(\theta), \ \text{ where } \ R(\theta) = \lambda \sum_{g=1}^{G} \sqrt{n_g} \ ||\theta_g||_2. \tag{3}$$

$R(\theta)$ is the $l_1/l_2$ penalty term which is the $l_1$ norm (promotes sparsity) of the $l_2$ norm (penalizes weights) over the non-overlapping parameter groups $(g)$, and $G$ is the total number of groups. Here $n_g$ is the group size, and $\lambda$ is the regularization parameter or strength or degree of regularization. The $l_2$ norm is given by $||\theta_g||_2 = \sqrt{\sum_{j=1}^{n_g} \theta_j^2}$. Since the penalty term is non-differentiable, in order to minimize the objective in Eq. 3, proximal gradient descent updates of the parameters are required,

as explained in [34]. The proximal updates are based on the gradients of the differentiable part of the composite loss function, i.e.,

$$\theta_{t+1} \leftarrow prox_{\alpha R}(\theta_t - \alpha \nabla_\theta L(\theta_t, D^{tr})), \tag{4}$$

where $prox_{\alpha R}$ is the proximal operator, and $\alpha$ is the learning rate. In spite of its non-differentiable nature, the proximal operator for the $l_1/l_2$ penalty term can be efficiently computed in a closed form [35], given by:

$$prox_{\alpha R}(\theta_g) = \begin{cases} \left[1 - \dfrac{\alpha\lambda\sqrt{n_g}}{||\theta_g||_2}\right]\theta_g & ; \ ||\theta_g||_2 > \alpha\lambda\sqrt{n_g} \\ 0 & ; \ ||\theta_g||_2 \le \alpha\lambda\sqrt{n_g} \end{cases}. \tag{5}$$

Given that the parameters are partitioned into $G$ dis-joint groups, the proximal operator likewise adheres to this decomposition and thus can be implemented on each parameter group. According to Eq. 5, if the norm of the parameter group is less than the strength of regularization (i.e., a function of regularization parameter $\lambda$, learning rate $\alpha$, and the number of elements in the group $n_g$), the entire group of parameters is set to zero resulting in a sparse solution.

**Proposed approach:** In this work, we apply group sparsity to the shared parameters $\theta_b$ of a multi-task network. Given that the foundation for MTL and group sparsity has been laid out (discussed above), we can now seamlessly delve into their integration. The parameter vector of a convolutional layer is a 4-D vector of dimension $(N, C, H, W)$, where $N$ is the number of filters, $C$ is the number of channels, $H$ and $W$ are the spatial height and width. The channel-wise structure sparsity is introduced in each convolutional layer of the backbone by considering each channel as a group, as demonstrated in [36]. Therefore, the multi-task optimization objective can now be written as (from Eq. 1 and 3):

$$\min_\theta \ L_{comb}(\theta, D^{tr}) + \lambda \sum_{g=1}^{G} \sqrt{n_g} \, ||\theta_{b_g}||_2 \tag{6}$$

where $||\theta_{b_g}||_2 = \sum_{l=1}^{L}\left[\sum_{n_c=1}^{C_l} ||\theta_b^l(:,n_c,:,:)||_2\right]$. Here $L$ is the total number of convolutional layers in the backbone network, and $C_l$ is the number of channels in the $l^{th}$ layer. In Eq. 6, the group lasso term i.e., $\lambda \sum_{g=1}^{G} \sqrt{n_g} \, ||\theta_{b_g}||_2$ zeroes out an entire channel group, i.e., $\theta_b^l(:,n_c,:,:)$ if its norm is below the degree of regularization as per Eq. 5, rendering the backbone parameters sparse. Therefore, $l_1$ sparsity leads to structured feature selection, while $l_2$ sparsity term promotes within-group regularization, ensuring that all features in that group are either jointly important or jointly unimportant. While the combined multi-task loss term i.e., $L_{comb}(\theta, D^{tr})$ fosters the joint learning of multiple tasks, such that the tasks help each other to learn better. The reduction of trainable parameters in a multi-task model through dynamic sparsification during its training phase can result in benefits during inference, such as decreased memory usage, computation requirements, and prediction time, as well as enhanced performance.

## 4. Experimental setup

**Datasets and Tasks:** We evaluate the concept of structured sparsity on multi-task learning on two publicly available datasets, i.e., the NYU-v2 dataset [37] and the CelebAMask-HQ dataset [38] (referred to as celebA dataset in this work). In the NYU-v2 dataset, which contains images of indoor scenes, three dense pixel-level tasks are chosen: semantic segmentation (comprising 40 classes), depth estimation, and surface normal estimation. For the celebA dataset, which is a large-scale face image dataset, two binary classification tasks of male/female and smile/no smile identification are considered. A pixel-level semantic segmentation task has also been selected, featuring three classes: skin, hair, and background. The celebA dataset is usually used in MTL to investigate task inter-dependencies.

**Network architecture and hyperparameters:** A three-channel RGB image of size $256 \times 256$ is fed to a dilated ResNet-50 [39] backbone network in batches of 16(32) for NYU-v2(celebA) dataset. The output of the backbone are shared representations which are then given to task-specific networks (Figure 2). A Deeplab-V3 network [40] is employed as the task head for the dense prediction tasks. While the task-specific network for classification tasks uses a minimal network with a convolution layer and two linear fully connected layers. This work employs cross-entropy loss for the semantic

segmentation task, inverse cosine similarity loss for surface normal estimation, means square loss for depth estimation, and binary-cross entropy loss for classification. Adaptive optimizer 'Adam' [41] is used for optimizing the non-differentiable objective function (Eq. 3) by calculating adaptive proximal gradients (explained in [34]). In order to ensure adequate training for all tasks, a learning rate $\alpha$ of 0.0001 is used. The regularization parameter $\lambda$ is a crucial hyper-parameter in this work, as it determines the degree of sparsity. Therefore, we conducted an ablation study considering a range of $\lambda$ values, i.e., $[1 \times 10^{-7}, 1 \times 10^{-6}, 1 \times 10^{-5}, 1 \times 10^{-4}, 1 \times 10^{-3}, 1 \times 10^{-2}]$. The dataset was divided into non-overlapping training, validation, and test set; for NYU-v2 dataset the same data split as [18, 42] is followed, while for the celebA dataset, we randomly split the data in 60-20-20%. A uniform test set and similar hyper-parameters are used to compare performance across all experiments. The training of the models is conducted on NVIDIA A100 Tensor Core GPUs, equipped with 40 GB of onboard HBM2 VRAM. To assess the consistency of the model, all experiments were conducted five times using distinct random seeds. The outcomes are presented in the form of the mean and standard deviation. The source code for reproducibility can be found at https://github.com/ricupa/Less-is-More-Towards-parsimonious-multi-task-models-using-structured-sparsity.git.

## 5. Results and discussion

To evaluate the performance of the proposed approach, two types of experiments were designed: single-task and multi-task experiments (mostly all possible task combinations are considered). Table 1 and Table 2 display the performance of the various single-task and multi-task experiments for the NYU-v2 and celebA datasets, respectively. Table 1 displays the results for three values of $\lambda$, i.e., 0 (no sparsity), $1 \times 10^{-6}$ & $1 \times 10^{-5}$, while Table 2 for $\lambda = 0$ (no sparsity) and $1 \times 10^{-5}$.

Table 1: Single task and multi-task (test set) performance for all the three tasks on the NYU-v2 dataset. The % group sparsity represents the fraction of groups that are eliminated after training. The values of %group sparsity and parameter sparsity are the mean values over 5 trials. Here IoU is the intersection over union, CS is the cosine similarity, and MAE is the mean absolute error; the ↑ represent a higher value is better and ↓ represent a lower value is preferable. The values denoted in *italics* indicate the optimal performance achieved across all tasks in a single-task configuration, incorporating group sparsity. On the contrary, the values highlighted in **bold** represent the top two performances attained in a multi-task setup, also utilizing group sparsity.

| | Lambda | Tasks | | | In the backbone network | |
|---|---|---|---|---|---|---|
| **Experiments** | $\lambda$ | **Segmentation** | **Surface Normal** | **Depth** | **% group** | **% parameters** |
| | | **IoU** ↑ | **CS** ↑ | **MAE** ↓ | **sparsity** | **reduced to zero** |
| Semantic | 0 | $0.3140 \pm 0.0350$ | | | 0 | 0 |
| Segmentation | $1 \times 10^{-6}$ | $0.3203 \pm 0.0096$ | | | 0 | 0 |
| | $1 \times 10^{-5}$ | *$0.3338 \pm 0.0068$* | | | 35.77 | 43.05 |
| Depth | 0 | | | $0.1645 \pm 0.0016$ | 0 | 0 |
| estimation | $1 \times 10^{-6}$ | | | *$0.1572 \pm 0.0017$* | 21.95 | 31.11 |
| | $1 \times 10^{-5}$ | | | $0.1625 \pm 0.0015$ | 89.90 | 93.57 |
| Surface normal | 0 | | $0.7077 \pm 0.0047$ | | 0 | 0 |
| estimation | $1 \times 10^{-6}$ | | *$0.7903 \pm 0.0041$* | | 17.87 | 20.42 |
| | $1 \times 10^{-5}$ | | $0.7699 \pm 0.0058$ | | 55.62 | 64.39 |
| segmentation + | 0 | $0.2233 \pm 0.0094$ | | $0.1656 \pm 0.0030$ | 0 | 0 |
| depth estimation | $1 \times 10^{-6}$ | $0.3308 \pm 0.0049$ | | $0.1379 \pm 0.0034$ | 0 | 0 |
| | $1 \times 10^{-5}$ | $0.3345 \pm 0.0073$ | | $0.1389 \pm 0.0027$ | 66.06 | 73.58 |
| segmentation + | 0 | $0.2276 \pm 0.0064$ | $0.7072 \pm 0.0087$ | | 0 | 0 |
| surface normal | $1 \times 10^{-6}$ | $0.3353 \pm 0.0082$ | $0.7797 \pm 0.0128$ | | 13.51 | 15.34 |
| estimation | $1 \times 10^{-5}$ | **$0.3480 \pm 0.0180$** | $0.7833 \pm 0.0058$ | | 64.05 | 72.97 |
| surface normal | 0 | | $0.6905 \pm 0.0037$ | $0.1719 \pm 0.0038$ | 0 | 0 |
| estimation + | $1 \times 10^{-6}$ | | $0.7609 \pm 0.0279$ | **$0.1299 \pm 0.0030$** | 60.62 | 69.83 |
| depth estimation | $1 \times 10^{-5}$ | | **$0.7967 \pm 0.0042$** | $0.1332 \pm 0.0015$ | 88.62 | 93.38 |
| segmentation + | 0 | $0.2171 \pm 0.0128$ | $0.6948 \pm 0.0080$ | $0.1657 \pm 0.0034$ | 0 | 0 |
| surface normal + | $1 \times 10^{-6}$ | **$0.3418 \pm 0.0062$** | $0.7814 \pm 0.0108$ | **$0.1301 \pm 0.0064$** | 21.80 | 23.85 |
| depth estimation | $1 \times 10^{-5}$ | $0.3394 \pm 0.0107$ | **$0.7857 \pm 0.0076$** | $0.1336 \pm 0.0029$ | 69.72 | 76.72 |

**Group sparsity enhances the performance:** For both datasets, the outcomes of the experiments show that even with approximately 70% sparsity, the multi-task models perform better than the non-sparse models; this is also true in the case of single-task settings. Upon assessing the task performance for the NYU-v2 dataset, as presented in Table 1, it is apparent that the implementation of group sparsity yields a notable improvement in the performance of all tasks, especially in the multi-task setting. For the semantic segmentation task, the best IoU score is achieved with a $\lambda$ value

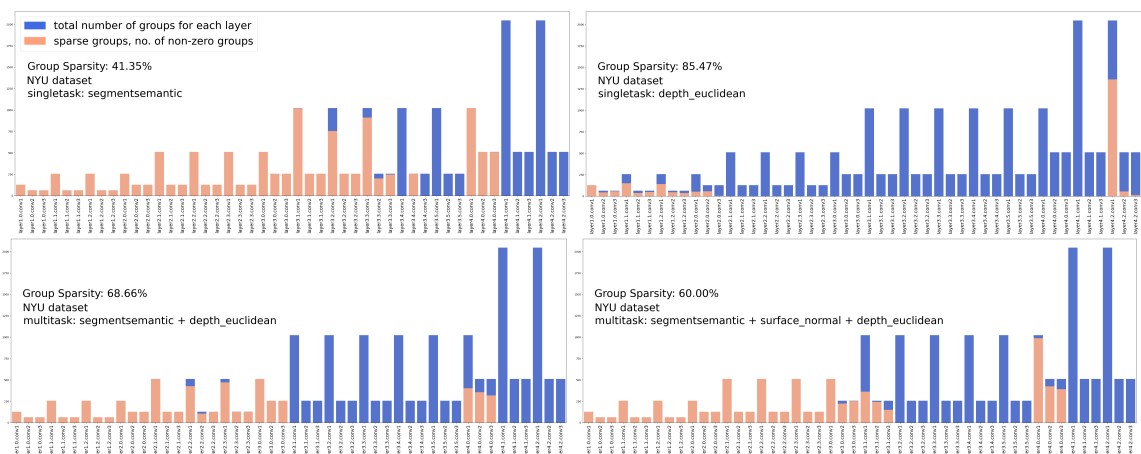

Figure 3: Comparison of the number of (non-zero) channels per convolution layer before (blue) and after sparsity (orange). The names of dilated ResNet-50 convolution layers are on the horizontal axis, and on the vertical axis are the number of channels. For all these plots, the value of $\lambda = 1 \times 10^{-5}$.

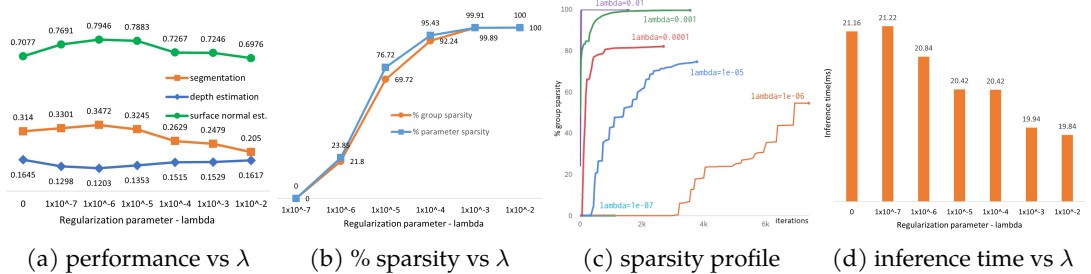

(a) performance vs $\lambda$  (b) % sparsity vs $\lambda$  (c) sparsity profile  (d) inference time vs $\lambda$

Figure 4: These figures demonstrate (a) the variations in the task's test performance, (b) group and parameter sparsity percentage, (c) sparsity profile during training, and (d) mean inference time for the multi-task scenario involving all three tasks of the NYU-v2 dataset, across a spectrum of regularization parameter values $\lambda$. The term "% group sparsity" refers to the proportion of eliminated groups, while "% parameter sparsity" indicates the ratio of parameters assigned a value of zero.

of $1 \times 10^{-5}$ in both single task ($0.3338 \pm 0.0068$) and the multi-task settings ($0.3480 \pm 0.0180$ in combination with surface normal estimation). The lowest MAE (mean absolute error) for the depth estimation task is achieved with a $\lambda$ value of $1 \times 10^{-6}$ ($0.1572 \pm 0.0017$) for single-task learning. In the multi-task setting, when depth estimation is combined with surface normal estimation, the performance improved significantly, with the lowest MAE being $0.1299 \pm 0.0030$ at a lambda of $1 \times 10^{-6}$. For surface normal estimation, in a single-task framework, the best result (CS) is for $\lambda = 1 \times 10^{-6}$ ($0.7903 \pm 0.0041$). However, the best performance throughout is when it is combined with depth estimation, specifically at a $\lambda = 1 \times 10^{-5}$ ($0.7967 \pm 0.0042$). It is evident that combining tasks can boost performance, but the effect is task-dependent. Segmentation performs well on its own or when supplemented with surface normal estimation, while depth estimation gains significantly from the surface normal estimation task (closely related tasks, with the latter often being derived from the former). In fact, combining depth estimation and surface normal estimation results in better performance and much higher group sparsity than when paired with semantic segmentation. In the absence of sparsity, i.e., $\lambda = 0$, the individual task performance surpasses that of any of the multi-task performances across all tasks. The reason behind this can probably be negative information transfer between tasks due to oversharing of information. The introduction of group sparsity in multi-task experiments yields a significant improvement in the performances of all tasks. Thus, it can be inferred that the implementation of group sparsity within the shared layers regulates the dissemination of information across tasks. Specifically, the utilization of $l_2$ regularization in the penalty term outlined in Equation 3 facilitates soft parameter sharing, while the application of $l_1$ regularization promotes sparse hard parameter sharing. Thus, the outcomes of the proposed approach show the effectiveness of group sparsity in MTL.

Table 2: Single task and multi-task (test set) performance for all the three tasks on the celebA dataset. The % group sparsity represents the fraction of groups that are eliminated after training. Here, IoU represents the metric intersection over union, and the ↑ represents that a higher value of the metric (IoU and accuracy) is better. The % group sparsity and % parameter sparsity (i.e., reduce to zero) contain the mean values obtained from five trials. The values highlighted in **bold** represent the top two performances attained in a multi-task setup, incorporating group sparsity.

| Experiments | Lambda | Tasks | | | In the backbone network | |
| | | Segmentation | classification | | % group sparsity | % parameters reduce to zero |
| | | | Male/female | smile/no smile | | |
| | | IoU ↑ | Accuracy ↑ | Accuracy ↑ | | |
| segmentation | 0 | $0.8885 \pm 0.0048$ | | | 0 | 0 |
| | $1 \times 10^{-5}$ | $0.9084 \pm 0.0033$ | | | 58.1 | 71.94 |
| male/female | 0 | | $0.6296 \pm 0.0019$ | | 0 | 0 |
| | $1 \times 10^{-5}$ | | $0.6284 \pm 0.0029$ | | 87.32 | 92.31 |
| smile/no smile | 0 | | | $0.5204 \pm 0.0026$ | 0 | 0 |
| | $1 \times 10^{-5}$ | | | $0.5238 \pm 0.0004$ | 100 | 100 |
| seg + male | 0 | $0.8363 \pm 0.0028$ | $0.9103 \pm 0.0217$ | | 0 | 0 |
| | $1 \times 10^{-5}$ | **0.9136± 0.0024** | **0.9726± 0.0035** | | 67.66 | 77.77 |
| male + smile | 0 | | $0.6284 \pm 0.0108$ | $0.5242 \pm 0.0006$ | 0 | 0 |
| | $1 \times 10^{-5}$ | | $0.7275 \pm 0.0016$ | **0.5308 ± 0.0103** | 63.18 | 78.1 |
| seg +male + smile | 0 | $0.8123 \pm 0.0024$ | $0.8641 \pm 0.0269$ | $0.5238 \pm 0.0012$ | 0 | 0 |
| | $1 \times 10^{-5}$ | **0.9040± 0.0035** | **0.8514± 0.1578** | **0.5241± 0.0760** | 77.1 | 83.41 |

Figure 3 illustrates the groups of parameters before and after pruning for all the convolution layers of the shared backbone network. Comparing single-task plots to multi-task plots shows how MTL facilitates learning shared features, and sparsity eliminates unimportant parameter groups. In multi-task model plots, initial layers are predominantly active, sharing low-level features with more parameters assigned to them than to later layers. Conversely, most intermediate layers are sparse, with only a few deep layers actively sharing high-level features. Such a sparsity pattern suggests that group lasso preserves vital network structures, such as residual connections of the ResNet backbone network.

Similar findings can also be asserted for the celebA dataset, as presented in Table 2. The tasks of segmentation and male/female classification exhibit significant enhancements with sparsity and in MTL setting. The accuracy of the smile/no smile classification task remains consistent (under-performing) even with the introduction of sparsity. As expected, combining image-level male/female classification with pixel-level semantic segmentation tasks significantly improves the performance of the classification task. When all the tasks are trained together, even with 77% sparsity, the tasks perform better or sometimes equivalent to their no sparse and single-task experiments. These results prove the significance of both MTL and group sparsity.

**% Sparsity-performance trade-off:** Figure 4a illustrates the average test performance of all three tasks in a multi-task setting when subjected to various regularization strengths (i.e., $\lambda$ on the horizontal axis). While Figure 4b displays the amount of % group sparsity and % parameter sparsity for different values of $\lambda$. It can be concluded from both these figures that performance improvements are observed as sparsity increases to a certain point, beyond which performance begins to decline. As the value of lambda rises, sparsity also increases, leading to improved task performance (i.e., IoU and CS ↑, while MAE ↓) up to $\lambda = 1 \times 10^{-5}$; beyond this point, any further increase in lambda, or consequently in % sparsity, results in declining performance. It is noteworthy that increasing $\lambda$ from $1 \times 10^{-6}$ to $1 \times 10^{-5}$ results in substantial growth in sparsity, while the performance metrics remain relatively stable i.e., no significant change. The model exhibits complete sparsity (i.e., all parameters groups are equal to zero) for higher values of $\lambda$ ($1 \times 10^{-3}$ and $1 \times 10^{-2}$), resulting in two implications. Firstly, the absence of shared parameters contradicts the notion of MTL. Secondly, the model is so simple (sparse) that it cannot acquire adequate knowledge, leading to underfitting. The present ablation study examined the impact of increasing regularization intensity on overall task performance. Therefore, there is a trade-off between task performance and the degree of sparsity.

**Faster inference time:** Model sparsity results in a simple model with fewer parameters for inference; therefore, an increase in sparsity decreases inference time (specifically the runtime for backbone only), as shown in Figure 4d. In this work, the mean inference time represents the time taken by the backbone or shared network to process a batch of data, determined over five experimental trials

using different batches of the same size to validate the robustness of the results. For $\lambda = 1 \times 10^{-6}$, optimal performance corresponds to approximately 21% group sparsity, reducing the inference time by 0.32ms compared to the dense network. In contrast, at $\lambda = 1 \times 10^{-5}$ with around 70% group sparsity, despite a slight dip in performance, the network's speed increases by 0.74ms. Therefore, the choice of $\lambda$ modulates the balance between model performance and inference speed.

**Dynamic sparsity:** In this work we apply dynamic sparsity that involves continuously adjusting which weights are pruned during training, while static sparsity is a fixed pruning of neural network weights after training. Figure 4c shows the sparsification profiles for different values of $\lambda$ while training all three tasks together for the NYU-v2 dataset. Smaller $\lambda$ values initially yield a dense model that gradually becomes sparse, improving performance. Conversely, larger $\lambda$ values lead to high sparsity from the start of training, limiting performance. Starting with all parameters and then progressively pruning them is akin to synaptic pruning in the human brain [7]). Just as the network begins with a rich set of dense parameters and sparsifies over time, the brain initially forms an excess of neural connections. As development proceeds, less vital synapses are eliminated, optimizing the brain for efficiency and environmental adaptation.

**Structured vs Unstructured sparsity:** We also compared the performance of $l_1/l_2$ (structured) sparsity with $l_1$ (unstructured) sparsity, the results are shown in Table 3. Although applying solely $l_1$ sparsity achieves similar levels of parameter sparsity, the performance with $l_1/l_2$ sparsity, as shown in Table 1, remains superior. Fine-tuning the regularization parameter ($\lambda$) could potentially lead to comparable or improved performance and even greater parameter sparsity. However, unstructured sparsity often faces challenges in hardware efficiency since it is a fine-grain approach that involves removing parameters randomly (without a pattern) based on some criteria. Conversely, while structured sparsity aligns better with hardware optimization, it typically faces limitations in achieving high levels of sparsity without adversely impacting performance because it is a coarse-grained approach for pruning structured blocks of parameters [7]. The choice between structured and unstructured sparsity depends on the use case, balancing between performance and hardware efficiency. Furthermore, as shown in Table 3, unstructured sparsity can also lead to the zeroing out of some channels, thus contributing to a notably low percentage of group sparsity.

Table 3: Performance of MTL with $l_1$ sparse regularization on the NYU-v2 Dataset. The regularization parameter ($\lambda$) is set at $1 \times 10^{-3}$, which yields a level of parameter sparsity comparable to that achieved with $l_1/l_2$ sparsity.

| MTL Experiments | Segmentation IoU ↑ | Surface normal est. CS ↑ | Depth est. MAE ↓ | % group sparsity | % parameter sparsity |
|---|---|---|---|---|---|
| Segmentation + Surface Normal est. | $0.2962 \pm 0.0050$ | $0.7421 \pm 0.0048$ | - | 0.59 | 74.31 |
| Depth est. + Surface Normal est. | - | $0.7285 \pm 0.0096$ | $0.1545 \pm 0.0011$ | 5.46 | 77.78 |
| Segmentation + Depth est. + Surface Normal est. | $0.2900 \pm 0.0258$ | $0.7389 \pm 0.0187$ | $0.1520 \pm 0.0052$ | 1.68 | 69.92 |

In general, the outcomes presented in this section for both datasets demonstrate the efficacy of group sparsity in MTL. The aforementioned approach is effective as it leverages the principle of inductive bias, wherein tasks in MTL share features and mutually reinforce learning. Additionally, the incorporation of $l_1/l_2$ group sparsity facilitates the removal of redundant parameters that do not contribute to any of the tasks while also regularizing the parameters. This approach yields a parsimonious backbone model with reduced parameters that accommodates all the tasks within a multi-task framework.

# 6. Conclusion and future scope

We present an approach for developing parsimonious models by employing dynamic group sparsity in a multi-task setting. Through extensive experiments, we demonstrate that sparse multi-task models perform as well as or better than their dense counterparts. So, sparsifying the model during training yields a more general model that offers faster inference times. Our proposed method can be integrated with any multi-task models that undergo gradient-based training. Furthermore, it is adaptable to various tasks, be they classification, regression, or otherwise. Therefore, we put forth a model-agnostic and task-agnostic approach for developing simple and interpretable multi-task models. In this study, the regularization factor ($\lambda$), which is a hyper-parameter, determines the level of sparsity. Finding the ideal value of $\lambda$ is a challenging task, though. Therefore, a promising future research direction might involve developing an approach for learning the optimal value of $\lambda$ while training, potentially leading to enhanced performance and optimal sparsity.

## Acknowledgements

The authors are grateful to the National Supercomputer Centre at Linköping University for their access to the Berzelius supercomputing system and acknowledge the generous support of the Knut and Alice Wallenberg Foundation.

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
