# OpenReview forum: "Less is More – Towards parsimonious multi-task models using structured sparsity"
_CPAL.cc/2024/Conference — CPAL 2024 (Proceedings Track) Oral_

### Official Review · Reviewer_sZcf · 2023-10-07

**Rating:** 6
**Confidence:** 5

**Review:**

This paper adopts the group lasso penalty for multi-task learning, resulting in cross-task structural sparsity.
The experimental results show that removing redundant parameters further improves MTL performance.
Moreover, this paper shows that appreciating sparsity not only reduces computation costs for each task but also benefits better task performance.

The proposed method is not novel but is effective in learning structured sparsity for multi-task learning, which is potentially utilized in various scenarios.
Considering this paper focuses on the sparsification of hard-shared parameters, the experiments are reasonable, but proper comparison is also helpful.
Like the MTL performance under other structure sparsity methods like SWP[1] and SSL[2].

[1] Pruning Filter in Filter

[2] Learning structured sparsity in deep neural networks

---

### Official Review · Reviewer_mbZZ · 2023-10-07

**Rating:** 6
**Confidence:** 4

**Review:**

The author introduces channel-wise l1/l2 group sparsity into the shared convolutional layer parameters (or weights) of multi-task learning models. However, lack of comparative analysis with other relevant sparse multitask learning methods limits its persuasiveness.

Pros:
+ Extensive experiments.

Cons:
+ Some paragraphs present unclear logic, such as: " .. Therefore, Sparsity aids in achieving parsimony ..." does not have a clear causal relationship with its preceding context.
+ Lacks comparative analysis with any other methods, such as [1]

[1] Sun T, Shao Y, Li X, et al. Learning sparse sharing architectures for multiple tasks[C]//Proceedings of the AAAI conference on artificial intelligence. 2020, 34(05): 8936-8943.

---

### Official Review · Reviewer_hye9 · 2023-10-15

**Rating:** 6
**Confidence:** 3

**Review:**

The paper proposes to learn structured (group) sparsity in MTL, i.e., it learns sparse shared features among multiple tasks. They analyzed the results of group sparsity in both single-task and multi-task settings on two widely-used Multi-Task Learning (MTL) datasets: NYU-v2 and CelebAMask HQ. On both datasets, which consist of three different computer vision tasks each, multi-task models with approximately 70% sparsity outperform their dense equivalents. I have following concerns:

1. The paper doesn't clearly explain how their approach bring benefit over many other sparse-learning works like ( https://arxiv.org/pdf/1911.05034.pdf https://arxiv.org/pdf/1705.04886.pdf). The novelty of the work is limited.

2. In addition, the motivation of the work in the introduction is not strongly enlisted (eg. organization of parameters in a CNN, CNNs can develop redundant filter). It will be important to explain why/how channel-wise l1/l2 penalty to the shared (CNN) layer parameters can help in solve complex computer vision tasks (contribution 1).

3. The paper lacks any comparable sparsity-induced MTL baselines in evaluation which is required to show the effectiveness. I am also curious to know how the authors calculated mean inference time in figure 4.

4. How do the authors regulate between Loss1, Loss2, ... Loss N of the proposed architecture in Figure 2. Seems very difficult to tune.

---

### Meta-Review · Area_Chair_aqca · 2023-11-12

**Recommendation:** Accept (Poster)
**Confidence:** 4

**Metareview:**

This paper proposes an approach that incorporates group sparsity into the shared CNN backbone in multi-task learning. The proposed approach is extensively evaluated on two datasets with heterogeneous CV tasks, showing that it can significantly prune the model parameters while having improved performance in comparison to non-sparse baselines.

During the rebuttal, the authors successfully clarified reviewers' questions about various technical details. There is still one major concern raised by all reviewers about the lack of comparison to various existing sparsity methods, which undermines the novelty of the proposed approach. However, reviewers agreed that the merits of this paper slightly overweigh its shortcoming.

---

### Decision · Program_Chairs · 2023-11-20

**Decision:**

Accept (Oral)

**Comment:**

The authors propose a method for learning structured sparsity in multi-task learning. Overall, the reviewers and AC agree that the approach proposed in the paper is an interesting contribution despite some missing comparisons and ablations. The method is evaluated on two widely-used multi-task learning datasets, showing that it can significantly prune the model parameters while having improved performance in comparison to non-sparse baselines. For the camera-ready version, the authors should strongly consider providing a more detailed explanation and experiments of how the proposed approach brings benefit over many other structured sparse-learning approaches.

The action PC chair for this paper is Gintare Karolina Dziugaite, who made the decision after carefully reading the paper as well as the comments by all reviewers and AC. The decision is agreed by all PC chairs.